# Biphasic Response of Astrocytic Brain-Derived Neurotrophic Factor Expression following Corticosterone Stimulation

**DOI:** 10.3390/biom12091322

**Published:** 2022-09-18

**Authors:** Alexandros Tsimpolis, Maria Kokkali, Aris Logothetis, Konstantinos Kalafatakis, Ioannis Charalampopoulos

**Affiliations:** 1Department of Pharmacology, Medical School, University of Crete, 71003 Heraklion, Greece; 2Institute of Molecular Biology and Biotechnology, Foundation for Research and Technology Hellas, 70013 Heraklion, Crete, Greece; 3Institute of Health Sciences Education, Barts and the London School of Medicine and Dentistry (Malta Campus), Queen Mary University of London, VCT 2520 Victoria, Gozo, Malta

**Keywords:** corticosterone, brain-derived neurotrophic factor, astrocytes, pulsatility, glucocorticoid receptors, TrkB, mineralocorticoid receptors

## Abstract

Novel research studies indicate multivarious interactions of glucocorticoid hormones (GCs) with the brain-derived neurotrophic factor (BDNF), regulating important aspects of neuronal cell physiology. While there is recent evidence of the chronic effects of GC stimulation on BDNF levels, as well as of the role of BDNF stimulation in the type of genomic effects following activation of GC-sensitive receptors, no data exist concerning the acute effects of GC stimulation on BDNF/TrkB gene expression. To address this question, we conducted a chrono-pharmacological study on rodent glial cells, astrocytes, which express the BDNF receptor, TrkB, following corticosterone administration. mRNA levels of BDNF and TrkB were estimated 1, 6, 12 and 24 h post-treatment. Selective inhibitors for GC-sensitive receptors and TrkB were used to decipher the molecular pathways of the effects observed. Our data support a biphasic response of BDNF expression after corticosterone stimulation. This response is characterized by a rapid TrkB phosphorylation-dependent upregulation of BDNF mRNA within the first hour, followed by a glucocorticoid receptor (GR)-dependent downregulation of BDNF mRNA, evident at 6, 12 and 24 h, with a direct impact on the protein levels of mature BDNF. Finally, a second pulse of corticosterone administration 1 h prior to the 6, 12 or 24 h timepoints normalized BDNF expression for the corresponding timepoint (i.e., mRNA levels became indifferent from baseline). These results present for the first time a biphasic regulation of the neurotrophin system based on glucocorticoid rhythmicity, further indicating complex trophic responses to temporal hormonal mechanisms in the brain microenvironment.

## 1. Introduction

Glucocorticoids (GCs) are steroid hormones with important immunomodulatory, metabolic and neurobehavioral effects, important for brain cell homeostasis and optimal neural network functionality [1]. GCs, aside from the circadian rhythm, are characterized by a more dynamic, underlying ultradian rhythm, in which hormonal pulses are periodically—multiple times daily—secreted by the adrenal glands into the bloodstream, creating an oscillatory pattern of plasma GCs, whose frequency under physiological, non-stressful conditions varies between mammalian species (about one hour for rodents, 2–3 times lower frequency for humans) [2]. This pulsatility is of potential neurobiological significance, since GCs intracellularly bind to and activate two types of sensitive receptors, glucocorticoid receptors (GRs) and mineralocorticoid receptors (MRs), which are known to be expressed in neural and glial cells in many brain regions [3,4,5]. Moreover, their binding properties with GCs and the post-binding molecular events caused, including their deactivation/cell (re)cycling fate, relate to GC concentrations [6]. In lower GC levels, the hormone is more effectively able to activate MRs over GRs and produce long-term genomic effects instead of rapid non-genomic effects. As GC levels increase, the hormonal binding to GRs becomes more efficient, and more rapid, non-genomic effects may emerge. Finally, GRs usually demonstrate a more dynamic profile of rapid associations and dissociations with the hormone, after which the receptor becomes recycled, and thus, for a certain amount of time, unavailable for further GC stimulation [7]. The importance of hormonal pulsatility has been demonstrated in various experiments; for instance, ultradian GC stimulation induces cyclic GR-mediated transcriptional regulation (gene pulsing) both in vivo and in vitro in rodents [8], while alterations in the frequency and duration of the pulses translate to changes in synaptic physiology [9,10] and behavioral responses to stressful stimuli [11,12,13,14].

Over the past 15 years, various sources of evidence indicate that the molecular biology of GCs interacts with that of the major neurotrophin in the central nervous system (CNS), the brain-derived neurotrophic factor (BDNF), regulating important aspects of neuronal cell physiology and impacting cellular viability, memory, emotional processing and stress adaptation [15,16]. These interactions are translated into cellular effects at multiple levels of regulation, from controlling the profile of gene expression [17] to mobilizing various second-messenger systems of signal transduction [18]. For instance, acute stimulation of GR (by GCs) facilitates glutamate release by directly interacting with the BDNF receptor, TrkB [19]. On the contrary, chronic exposure to GCs (resulting in a downregulation of the GR expression) has opposite effects [20]. These studies have introduced a significant scientific field concerning the long-standing questions on the effects of hormonal microenvironment on trophic support in the brain under physiological and pathological conditions. It is of note that GCs and their receptors are considered the major regulating factor in stress, a condition where BDNF is also highly involved [21,22].

In this context, we aimed at exploring whether and how a GC pulse affects BDNF expression levels at various timepoints throughout a 24-hour period. We chose to focus our study on a specific glial type of cells, astrocytes, which express the BDNF receptor, TrkB, while they produce and secrete all neurotrophins. In addition, astrocytes present characteristics of immune cells since they are activated upon local immune activity, thus providing a suitable substrate for GC-mediated immuno-responses and exerting a significant role in many neuropathological conditions [23].

In the present research study, we performed a series of in vitro experiments, focusing on the temporal changes in BDNF and TrkB expression, and if and how the profile of BDNF expression changes when we introduce a second GC pulse. Conclusively, we now show that Cort administration has a two-phase response in astrocytic *Bdnf* gene expression, increasing its levels at 1 h but decreasing its expression at 6, 12 and 24 h after Cort treatment. In addition, the mRNA substrate decrease in the latter response leads to the subsequent reduction in the protein levels of mature BDNF (mBDNF). TrkB receptor seems to mediate the early response of GCs in Bdnf expression due to its phosphorylation/activation by Cort administration, in similar levels to BDNF ligands, while the late inhibitory response of GCs to Bdnf is mediated by the GR (and not MR) activation. It is of special interest that a short (1 h) priming pulse of Cort at all late treatments fully abolished the inhibitory effects, indicating for the first time—to our knowledge—that pulsatility of GCs could significantly impact BDNF production and thus strongly regulate basic functions of the nervous system, such as neuroprotection, neuro-/glio-genesis and synaptic enforcement, and thus modulate behavioral outcomes such as stress response, cognition and memory.

## 2. Materials and Methods

### 2.1. Reagents

Corticosterone (27840), RU-486 (M8046), spironolactone (S3378) and Ara-C (C1768) were obtained from Sigma-Aldrich, St. Louis, MO, USA. BDNF (450-02) was obtained from Peprotech, London, UK.

### 2.2. Primary Astrocytic Cultures

Mixed glial cultures were isolated from cortices of C57B/6 pups at post-natal day 2 (P2). P2 pups were decapitated and their cortices were isolated and kept in cold 1X HBSS. Cortices were enzymatically (using trypsin 1.25 μL/mL for 10′) and then mechanically dissociated to generate a single-cell suspension, which was then transferred to T75 flasks for cultivation. Approximately 2–2.5 cortices were used for each T75 flask. Mixed glial cultures were cultivated in a defined cortex medium containing high-glucose DMEM, 200 U mL^−1^ penicillin, 200 μg mL^−1^ streptomycin and 10% fetal bovine serum (FBS), until confluency in the flask was achieved (7–8 days). Then, medium in the flask was exchanged for fresh medium containing the anti-mitotic agent Ara-C at a final concentration of 10 μM, and the culture was incubated for another 3–4 days in order to effectively target the highly proliferative microglial cells. Ara-C was then removed from the flask and astrocytes were split and seeded for the experiment the next day. The astrocytic cultures generated from this protocol have >97% purity. Twenty-four hours prior to the beginning of the experiment, medium in the cultures was exchanged to fresh medium without FBS to ensure total deprivation of corticosterone. Baseline expression levels of *NR3C1* (gene that encodes for GR) and *NR3C2* (gene that encodes for MR) in the cultures were evaluated, with the expression of the former receptor being favored by ~4 times (Appendix A).

### 2.3. Treatments and Primary Outcomes

To investigate the main effects of Corticosterone (Cort) on BDNF expression in astrocytes, astrocytic cultures prepared as described above were treated with Cort (at a concentration of 225 nM, which imitates the physiological murine concentration) for either 1 h, 6 h, 12 h or 24 h. To distinguish the implication level of GR, MR and TrkB receptors in the transcriptional alterations mediated by Cort administration, cultures were treated with either a combination of Cort (225 nM) and a selective inhibitor of each of the aforementioned receptors, i.e., RU-486 (2.5 μM, 10 μM and 25 μM), spironolactone (2.5 μM) and ANA-12 (1 μM), for a total of 1 h, 6 h, 12 h or 24 h. To examine the effects of a second Cort pulse on BDNF homeostasis, astrocytic cultures were treated with medium containing Cort (225 nM) for either 5 h, 11 h or 23 h before being quickly washed and refilled with fresh medium containing Cort (225 nM) for 1 more hour. For each timepoint in the aforementioned experiments, we measured the expression of the *Per1* gene as a well-established positive marker of the activation of the GR transcriptional machinery due to Cort administration. The expression patterns of *Bdnf* and *Ntrk2* genes were also evaluated to determine the potential effects of Cort administration on BDNF expression and its high-affinity receptor, TrkB, respectively.

To demonstrate the Cort-mediated phosphorylation and activation of the TrkB receptor, astrocytic cultures were treated with Cort (225 nM) for either 20′, 40′ or 60′ and with BDNF (500 ng/mL), with 20′ as a positive control of TrkB phosphorylation. TrkB activation was evaluated as the ratio of the phosphorylated TrkB protein to the total TrkB protein in each condition.

### 2.4. qRT-PCR

Total RNA was extracted from cultures using TRIzol Reagent (Thermo Fisher, 15596026), and cDNA was synthesized using the High-Capacity cDNA Reverse Transcription kit (Thermo Fisher, Waltham, MA, USA, 4368814) according to the supplier protocols. Primers were designed using the NCBI Primer BLAST software [24]. We selected primer pairs with the least probability to amplify non-specific products (as predicted by the NCBI primer BLAST) and high ΔG values to avoid self or pair dimers and hairpin formations (using PrimerSelect by DNASTAR). All primers had 95–105% efficiency. Primer pairs were specifically designed to amplify products that span an exon–exon junction in order to avoid the amplification of genomic DNA. The primer pair for *Bdnf* was designed to detect an area inside the coding sequence (CDS) of the gene, thus being able to amplify all possible splicing variants expressed. To run the quantitative RT-PCR, we used 1 μL of cDNA (10 ng/μL) and the KAPA SYBR Fast kit (Sigma-Aldrich, St. Louis, MO, USA, KK4601) according to the supplier’s instructions. The cycling program consisted of 20 s at 95 °C, followed by 40 cycles of 95 °C for 3 s and 60 °C for 30 s on a StepOne Real-Time PCR System (Thermo Fisher Scientific, Waltham, MA, USA). After the completion of qPCR, a melt curve of the amplified products was performed. The housekeeping gene “Actin” was used to normalize the expression levels between the different conditions. Data were collected and analyzed using the StepOne Software v2.3 (Thermo Fischer Scientific, Waltham, MA, USA). Mouse primer sequences used are listed in Appendix A.

### 2.5. Western Blot Analysis (WB)

Protein samples were collected at 4 °C in Pierce IP Lysis buffer (87788, Thermo Fisher, Waltham, MA, USA) containing protease inhibitor cocktail (539131, Merck Millipore, ) and phosphatase inhibitor cocktail (524629, Merck Millipore, KGaA, Darmstadt, Germany). The total protein concentration of the samples was calculated using the BCA assay (23277, Thermo Fisher Scientific, Waltham, MA, USA), and equal amounts of total protein were loaded onto 8% Tris-HCl gels. After electrophoresis (110 V for 1 h), proteins were transferred to nitrocellulose membranes (GE10600002, Amersham Protran, Sigma-Aldrich, St. Louis, MO, USA). Blots were probed overnight at 4 °C with 1:1000 anti-TrkB (07-225-I, Merck Millipore, KGaA, Darmstadt, Germany), 1:1000 anti-phospho-TrkB (ABN1381, Merck Millipore, KGaA, Darmstadt, Germany) or 1:5000 anti-GAPDH (G8795, Sigma-Aldrich, St. Louis, MO, USA). The next day, blots were incubated with 1:5000 HRP-conjugated secondary antibodies for 1 h at room temperature and developed using SuperSignal West Pico PLUS Chemiluminescent Substrate (34580, Thermo Fisher Scientific, Waltham, MA, USA). Total TrkB and phospho-TrkB (pTrkB) expression were measured in different Western blots, normalized in each case to GAPDH, and the pTrkB/TrkB ratio was calculated.

### 2.6. Immunocytochemistry

Glass coverslips confluent on astrocytes were quickly washed with cold PBS and fixed for 10′ with 4% paraformaldehyde. The coverslips were then probed overnight at 4 °C with 1:100 anti-phospho-TrkB (ABN1381, Millipore) and 1:1000 anti-GFAP (Glial Fibrillary Acidic Protein), a well-established astrocytic marker (ab5541, Merck Millipore, KGaA, Darmstadt, Germany). Primary antibodies were visualized the next day with suitable secondary antibodies conjugated with fluorophores (Invitrogen, Thermo Fisher Scientific, Waltham, MA, USA). Coverslips were then mounted on slides and imaged using a Leica DMi8 inverted confocal microscopy in order to obtain around 20 consecutive optical sections (0.3 μm interval thickness). Obtained stacked images were then processed using ImageJ to properly subtract background noise from all channels.

### 2.7. ELISA

Protein samples or total culture medium were collected as previously described in Section 2.5 for the WB analysis, and their protein concentration was calculated using the BCA assay so that they would all be diluted down to the same concentration. mBDNF protein levels were measured using a BDNF ELISA kit (SK00752-03, Aviscera Biosciences, Santa Clara, CA, US) following the manufacturer’s protocol. Briefly, 50 μL of astrocytic lysate samples, medium or standards were added to an anti-BDNF pre-coated 96-well plate for 2 h at room temperature. The plate was then thoroughly washed and incubated for 90′ with the included Antibody Detection kit. The plate was again washed thoroughly and incubated for 45′ with streptavidin–HRP conjugate before the addition of the substrate solution for the development of the signal. The absorbance was recorded at 450 nm on a plate reader (TECAN Infinite M200 Pro). The amount of mBDNF in pg/mL was calculated by comparing their O.D. to the standard curve prepared in the same plate.

### 2.8. Statistical Analysis

Quantified data are presented as mean ± S.E.M. All statistical analyses were performed using GraphPad Prism 8 software. Data were analyzed by one-way ANOVA, followed by Dunnett’s multiple post hoc test for comparing more than three samples.

## 3. Results

### 3.1. Cort Stimulation Regulates a Temporally Distinct Two-Stage Response with Opposite Effects on BDNF Expression in Astrocytes

Initially, we wanted to establish the 24-hour-long baseline pattern of *Bdnf* expression profile on our astrocytes after the administration of Corticosterone (Cort) at concentration levels that would imitate the secretion of a physiological GC pulse in murine. Taking into account measurements of the circulating corticosterone levels in the serum and plasma of unstressed mice [25,26], we determined that the preferable concentration is at 225 nM. In order to ensure that we would be able to detect both early and late effects of the Cort administration, we established four timepoints (1 h, 6 h, 12 h and 24 h) to measure *Bdnf* expression levels with qRT-PCR and compared them to the respective expression levels of cultures treated with the appropriate concentration of vehicle (Figure 1a).

Using this experimental design, we firstly examined the ability of astrocyte cultures to recognize and react to the administration of the aforementioned Cort concentration by measuring the expression levels of the *Per1* gene. *Per1* is a circadian rhythm gene whose transcriptional regulation is directly associated with GC secretion [27]. Astrocytes were indeed able to respond to Cort administration, as indicated by the significant upregulation of the *Per1* gene, even after 24 h, with a slight tendency of reducing its levels as the day progressed (Figure 1b).

Having demonstrated that the selected concentration of Cort can initiate the activation of the GR transcriptional machinery, we sought to measure its effects on the expression of *Bdnf* in astrocytes. Interestingly, we were able to characterize the existence of two temporally distinct responses with opposing outcomes (Figure 1c). The early response to Cort administration is fast and occurs within the first hour. During this early response, *Bdnf* expression is significantly upregulated, almost by twofold compared to the control. After 6 h of treatment with the physiological Cort administration, the second and later responses follow, exerting an opposite effect compared to the early response, by significantly downregulating *Bdnf* expression levels. This late inhibitory effect lasted for the whole remainder of the 24 h treatment. These results suggest the existence of a differentially mediated mechanism sensitive to GC alterations. It should be noted here that there was no statistically significant alteration in the expression levels of the TrkB receptor gene, *Ntrk2* (Figure 1d), indicating a ligand-selective effect. Since it is well known that astrocytes are the main suppliers of neurotrophin load in the nervous system, these observations depict a strong pleiotropic connection of GCs and neurotrophin BDNF that could significantly regulate the homeostatic response of the tissue to hormonal changes.

In order to evaluate whether BDNF protein levels follow the same biphasic fluctuation over time after Cort administration that was observed in the mRNA levels, we measured the mBDNF protein levels on astrocytic lysates or total cell medium at the same timepoints using ELISA (Figure 1e). Firstly, we did not detect any BDNF protein in total medium, probably due to a lack of factors that induce its release or due to its small concentration in the total medium. However, we were able to measure the intracellular BDNF protein component. Interestingly, we did not detect any upregulation within the first hour after Cort administration in the intracellular protein levels of mBDNF. However, over time, a downregulation trend appears as soon as 6 h and significantly progresses after 12 and 24 h, extending to a maximum decrease of around 30% compared to the untreated control astrocyte cultures. This mBDNF downregulation correlates to the previously characterized reduction in the *Bdnf* mRNA substrate, further reinforcing the well-characterized inhibitory effects of chronic GC stimulation on BDNF expression.

### 3.2. GR Mediates the Late Response Inhibiting Bdnf Expression

The next step was attempting to decipher the receptors involved in the regulation of this mechanism. Since the observed responses were mediated by the administration of Cort, the first candidate receptors whose implications we investigated were the GC-sensitive receptors, GR and MR.

In order to identify each receptor’s participation in GC effects, we utilized the potent chemical inhibitors mifepristone or RU-486 and spironolactone (for GR and MR, respectively). The administration of each inhibitor was performed simultaneously with the administration of Cort, and the experimental design that was followed was exactly the same as the one described in Section 3.1. The effectiveness of RU-486 in inhibiting the transcriptional effects of GR in three different concentrations (2.5 μM, 10 μM and 25 μM) was certified by measuring the mRNA levels of the *Per1* gene, which were found to be significantly diminished in all three concentrations (Figure 2a, graph shown only for 25 μM of RU-486). Spironolactone, as expected, had absolutely no effect on the expression levels of *Per1* (Figure 2a), since the regulation of *Per1* expression is a hallmark of the activation of the GR and not the MR transcriptional machinery. By administering RU-486 in different concentrations, we were able to detect a dose-dependent reversion of the late-inhibitory response on *Bdnf* expression, with lower concentrations (2.5 μM) partially ablating the Cort-mediated suppression of *Bdnf*, while higher concentrations (25 μM) completely reversed this effect, leading to *Bdnf* overexpression of up to 3-fold compared to the control condition 6 h after the treatment (Figure 2b). No statistically significant alteration was observed in the early response, indicating that GR does not participate in the regulation of *Bdnf* expression in a rapid manner after Cort administration. On the other hand, spironolactone administration demonstrated a small, non-significant decrease (~45%) in the overexpression of *Bdnf* observed during the early response and no involvement at all of MR in the late response (Figure 2c).

### 3.3. Astrocytic TrkB Receptor Is Necessary for the Regulation of BDNF Overexpression in the Early Response to Cort Treatment

Pioneering studies by Jeanneteau et al. [28] have demonstrated the ability of GCs to activate the BDNF receptor (TrkB). Based on these observations, we sought to investigate whether activation of the TrkB receptor could be implicated in the mechanism that regulates the early response to Cort administration, thus creating a positive feedback loop of the BDNF–TrkB system in these glial cells.

To enlighten this hypothesis, we performed the same experiment, administering Cort as described in Section 3.1 for the same timepoints, with the addition of the TrkB selective inhibitor, ANA-12, at a concentration of 1 μM. Once again, the administration of the inhibitor occurred simultaneously with Cort. ANA-12 was able to almost fully perish the early response of *Bdnf* overexpression, as shown in Figure 3a, while at the same time had no effect on the late inhibitory response (6 h, 12 h and 24 h).

The implication of the TrkB receptor in the cellular mechanism that mediates the early response of GCs in astrocytes is further corroborated by the significant increase in the levels of phosphorylated TrkB only 20 min after the administration of a physiological Cort concentration (225 nM). Interestingly, the activation of TrkB at this timepoint could be even higher than the activation mediated by the administration of its native ligand (*p*-value = 0.08), BDNF (500 ng/mL), which was used as a positive control of TrkB activation (Figure 3b,c). Interestingly, it remains highly phosphorylated even 40 min after the introduction of Cort to the culture, with a slight decrease after 1 h.

### 3.4. A Second Cort Pulse Normalizes Bdnf Expression in Astrocytes

GC secretion, both in humans and mice, naturally occurs in periodic oscillations throughout the day. This pulsatile effect has been strongly associated with many homeostatic events all over the human body, including the brain [2]. Thus, it was hypothesized that GC pulsatility could mediate similar properties over *Bdnf* expression.

To address this, we introduced to our experimental design on astrocytes, a second administration of the same physiological Cort concentration (225 nM) exactly 1 h before the end of each of the 6 h, 12 h and 24 h treatments (at 5 h, 11 h and 23 h, respectively), as illustrated in Figure 4a. Initially, we evaluated the effects of the second pulse on the expression of the control gene, *Per1* (Figure 4b). The second administration of Cort elevated its expression in all three timepoints compared to the respective timepoints in Figure 1b at the exact same level, while abolishing, at the same time, the reductive tendency on gene expression that was initially observed at the later timepoints.

For *Bdnf*, the obtained results from the second administration (GC pulse) confirmed our initial hypothesis, demonstrating a full restoration of its gene transcription levels back to the control (untreated condition) levels at all of the tested timepoints (Figure 4c). The significance of these results is not limited only to the measured expression values, but also to the broad timeline in which the astrocytes can “remember” the initial Cort stimulation and regulate their *Bdnf* expression levels accordingly, so as to maintain a healthy homeostatic profile.

## 4. Discussion

The notion that GC stimulation creates a contextual framework in which other biomolecules with GC-sensitive downstream pathways may exert differential effects could be studied in the case of BDNF, which is the major neuroprotective, neurogenic and synaptogenic factor of the central nervous system. To interpret such complex interactions, though, we need to initially identify the potential effects of GC stimulation on the BDNF/TrkB system. Recent evidence indicates that exposure of rats to high corticosterone levels for over a month led to changes in the protein levels of the prodrome form of BDNF in the hippocampus, cerebellum, pituitary glands (increased compared to controls) and adrenal glands (decreased compared to controls), and these changes were correlated to depressive-like behavior [29]. Moreover, the effects of BDNF stimulation on the conditioning of GC-induced modes of GR function (as a transcription factor) has also been shown [30]. In our study, on the contrary, we aimed at exploring the potential acute, short-term dynamics of *Bdnf* gene expression in a time-dependent manner, from 1 h to 1 d after corticosterone stimulation.

We chose to focus on glial cells, specifically the astrocyte population, as our cells of interest for this in vitro experiment, acknowledging the rapidly increasing literature indicating their crucial role in brain physiology and pathology [31], including processes such as cognition, emotional processing and neural network homeostasis that were once considered to be an almost exclusive aspect of neuronal function [32]. It is worth noting that although our in vitro cell culture system consists of pure astrocytic populations, thus ensuring that the represented results are indeed astrocyte-specific, they do have the limitations of a 2D in vitro cell culture system, lacking their physiological characteristics’ in vivo morphology and cell-to-cell interaction, as well as the important interplay with other glial and neuronal populations. For this reason, our results should be not only replicated in neuronal cell cultures, but also in mixed neuronal–glial cultures, as well as in human cell cultures to elucidate whether our findings are preserved or change under the effect of different or more complex biological systems. The latter studies are under investigation in our current research projects. If these preliminary research results are confirmed in human models of astrocyte–neuronal populations, under physiological or increased, pulsatile-mediated concentrations of glucocorticoids, this could reveal new therapeutic windows for the use of the glucocorticoids and their analogs against major neuropathological conditions such as depression, chronic stress or even neurodegenerative diseases.

In the type of brain cells we chose to study, *NR3C1* (GR) mRNA levels were 4 times higher compared to *NR3C2* (MR) levels, and corticosterone stimulation provoked a strong GR-dependent response of *Per1* gene expression that lasted throughout the 24 h. Thus, our astrocytic cell cultures were confirmed to be GC-responsive.

While corticosterone administration does not seem to affect *Ntrk2* (TrkB) gene expression, it does modulate *Bdnf* mRNA levels in a biphasic manner. Within the first hour after corticosterone administration, *Bdnf* mRNA levels increased, a phenomenon that was not affected by blocking MRs or GRs with spironolactone or RU-486, respectively. This phenomenon, though, was preceded by a significant increase in TrkB phosphorylation (reaching a maximum at around 20 min) and was abolished when using a TrkB inhibitor, ANA-12, indicating that TrkB phosphorylation is involved in this early increase in *Bdnf* mRNA levels following corticosterone administration. The fact that this early effect of corticosterone does not implicate neither MRs (able to be suppressed by spironolactone) nor GRs (able to be suppressed by RU-486) is not necessarily surprising. Even if further studies are required to elaborate on the underlying molecular biology of the phenomenon, it could be rationalized by the inability of the GR/MR system to mediate rapid cellular responses, since it demands transcriptional effects. Recent evidence of membrane-initiated rapid signaling properties of several steroids could explain part of these results.

After the first hour, the pattern of *Bdnf* gene expression following corticosterone administration is reversed, and *Bdnf* mRNA levels become downregulated, an effect that is evident at 6 and 12 h and less evident at 24 h. This effect seems to be GR-dependent, as RU-486 (GR inhibitor) administration at gradually increasing doses eliminates (2.5 μΜ and more consistently at 10 μM) or even reverses (25 μM) the effect. It is worth noting that this late effect of GCs is more consistent with multiple reports on the capacity of GCs (especially in the context of chronic exposure or chronic stress) to downregulate BDNF expression [33]. Future studies may also aim at giving a mechanistic explanation of why RU-486 (mifepristone) in higher concentrations not only eliminates the late effect of GCs on BDNF expression, but also reverses it (i.e., *Bdnf* mRNA levels increase, as during the first hour following corticosterone administration, but at higher values). A possible explanation could be that at such concentrations, mifepristone might effectively bind to other target receptors (such as sex hormone receptors) and exert parallel pharmacological effects.

Following the aforementioned results at the transcriptional level, we sought to investigate their translatability to the mature form of the BDNF protein. Using ELISA, we measured the protein levels of mBDNF on the lysates of astrocytes at the same timepoints after corticosterone administration as the ones performed on the mRNA levels. Although our results at the later timepoints (12 and 24 h) were consistent with their respective ones at the mRNA level, indicating a gradual, statistically significant decrease in the protein levels (about 30% after 24 h), the measurements at the earliest timepoint (1 h) failed to reveal an increase in the mBDNF protein, which would agree with the previously described early response results, but rather remained unchanged compared to the untreated control condition. This outcome does not necessarily imply that corticosterone-mediated *Bdnf* overexpression after 1 h does not translate to mBDNF increase in astrocytes. Our inability to detect an increase in mBDNF levels could also be attributed to the experimental design, as 1 h could be considered limited time for a glucocorticoid-mediated genomic effect to become detectable in the translational level, or to the repletion of internal astrocytic BDNF stores that would counterbalance the production of superfluous mBDNF. Future studies should attempt to investigate this hypothesis by instigating the release of BDNF vesicles from astrocytes (e.g., with KCL) prior to the administration of corticosterone and measuring the mBDNF levels at later timepoints (e.g., 2 or 3 h). On the other hand, the direct correlation of the mRNA substrate diminution after extended corticosterone presence (12 and 24 h) to the significant decrease in mBDNF protein in astrocytes is consistent with studies describing reduced BDNF levels in patients with major depressive disorder [34], as well as in animal models of depression [35,36].

Finally, a second pulse of corticosterone 1 h prior the 6, 12 or 24 h timepoint normalized BDNF expression for the corresponding timepoint, i.e., the *Bdnf* mRNA levels for that timepoint did not differ from the 0 timepoint. Future studies are required to elaborate on the molecular mechanisms underlying this effect. It is worth noting, though, that different experimental settings [10] have illustrated the importance of periodic events of GC stimulation in establishing homeostasis in GC-sensitive mammalian neurobiological systems.

Our work has laid the foundation for investigating the effects of corticosterone stimulation on the BDNF expression levels of astrocytes, characterizing, for the first time, the existence of a biphasic response, implicating different molecular pathways for the regulation of each response. However, many questions arose and require further investigation. Future studies should try to elaborate the exact mechanism of early response characterized here and the receptors implicated in it, in an attempt to further investigate the potential neuroprotective properties that the significant, rapid increase in *Bdnf* mRNA levels of astrocytes after corticosterone stimulation might exert. In addition, the effects of corticosterone on the BDNF expression should be explored in other neuronal and glial populations, as well as in co-cultures of them, in in vitro configurations that would better imitate the physiological morphology and interaction between them (e.g., 3D cultures).

## 5. Conclusions

In this work, we have provided evidence of a biphasic response of BDNF expression after corticosterone stimulation in astrocytic cultures. This response is characterized by a rapid upregulation of *Bdnf* mRNA within the first hour, followed by a downregulation of *Bdnf* mRNA, evident at 6, 12 and 24 h, which subsequently led to a significant reduction in the levels of mBDNF. While our results show that there is no direct involvement of any of the two GC-sensitive receptors (MRs, GRs) in the rapid effect of corticosterone, the latter was dependent on a significant increase in the phosphorylation/activation of the TrkB receptor (reaching a maximum at 20 min). Furthermore, we were able to establish the direct involvement of GR in the late effects of corticosterone, as RU-486 (GR inhibitor) administration at gradually increasing doses eliminated or even reversed that effect. Finally, and perhaps more interestingly, a second pulse of corticosterone 1 h prior to the 6, 12 or 24 h timepoints normalized BDNF expression for the corresponding timepoint.

## Figures and Tables

**Figure 1 biomolecules-12-01322-f001:**
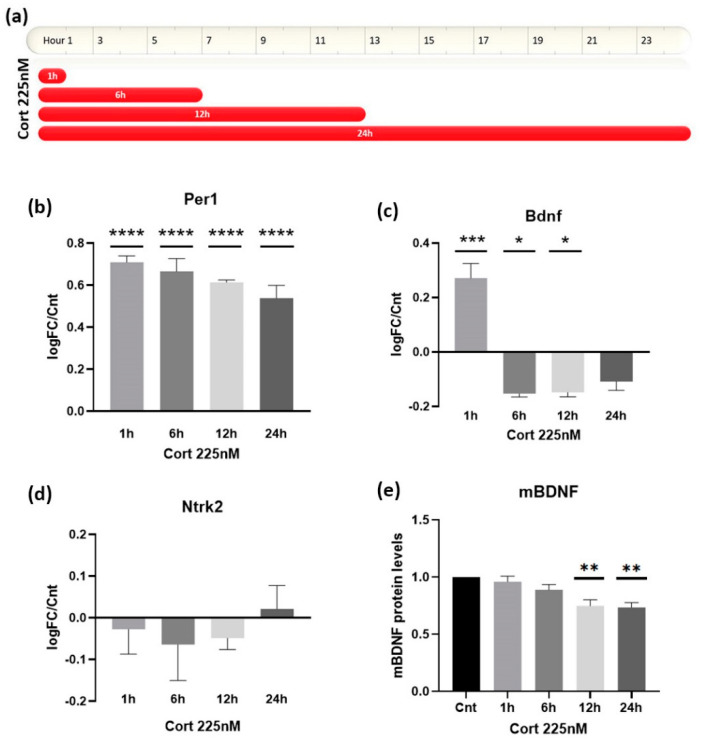
(**a**) Schematic illustration of the experimental design that was used to study the short-term and long-term transcriptional effects of Cort on *Bdnf* in astrocytes. (**b**) Cort administration (225 nM) strongly upregulates *Per1* mRNA expression levels on astrocytes, with effects lasting up to 24 h. qRT-PCR was performed in RNA samples collected after each timepoint, and the results are expressed as the logarithmic fold change (logFC) normalized to the untreated control. (**c**) Cort on astrocytes mediates a two-stage response in the expression of *Bdnf*. Rapidly (1 h) after Cort administration, the mRNA levels of *Bdnf* start to elevate significantly, while later, a persistent inhibitory effect follows that can last for up to 24 h. (**d**) Cort administration (225 nM) had no effect on the expression of the mRNA levels of the TrkB gene, *Ntrk2*, on astrocytes. (**e**) ELISA measurements of mBDNF protein levels on astrocytes reveal no change 1 h after Cort stimulation but a statistically significant decrease after 12 h and 24 h. The results from the ELISA measurements are represented as normalized to the untreated control condition. *n* = 3 for each experiment. * *p* < 0.05, ** *p* < 0.01, *** *p* < 0.001, **** *p* < 0.0001; one-way ANOVA. Data are mean ± S.E.M.

**Figure 2 biomolecules-12-01322-f002:**
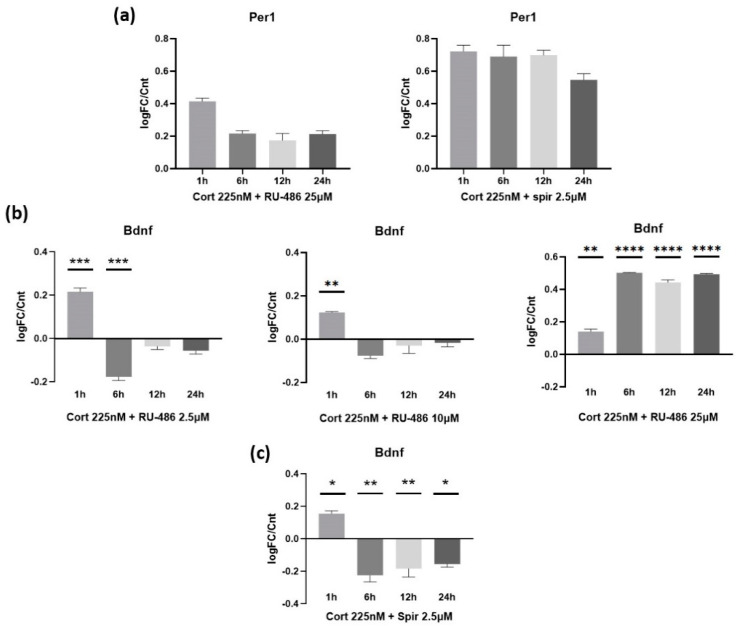
(**a**) The mRNA expression levels of *Per1* gene were measured after the administration of RU-486 25 μM (left) and spironolactone 2.5 μM (right) along with Cort (225 nM). RU-486 markedly inhibited the expression of *Per1*, while spironolactone had no effect whatsoever, as expected. (**b**) RU-486 was able to diminish the late inhibitory effect of Cort administration in a dose-dependent manner, with higher concentrations (25 μM) even allowing for complete reversion of the effect, leading to *Bdnf* overexpression for up to at least 24 h. (**c**) The effects of spironolactone (2.5 μM) on *Bdnf* expression in astrocytes after Cort administration (225 nM) were evaluated on astrocytes at 4 timepoints (1 h, 6 h, 12 h, 24 h). Spironolactone did not demonstrate any significant effect on neither the early nor the late response mediated by the Cort administration. *n* = 3 for each experiment. * *p* < 0.05, ** *p* < 0.01, *** *p* < 0.001, **** *p* < 0.0001; one-way ANOVA. Data are mean ± S.E.M.

**Figure 3 biomolecules-12-01322-f003:**
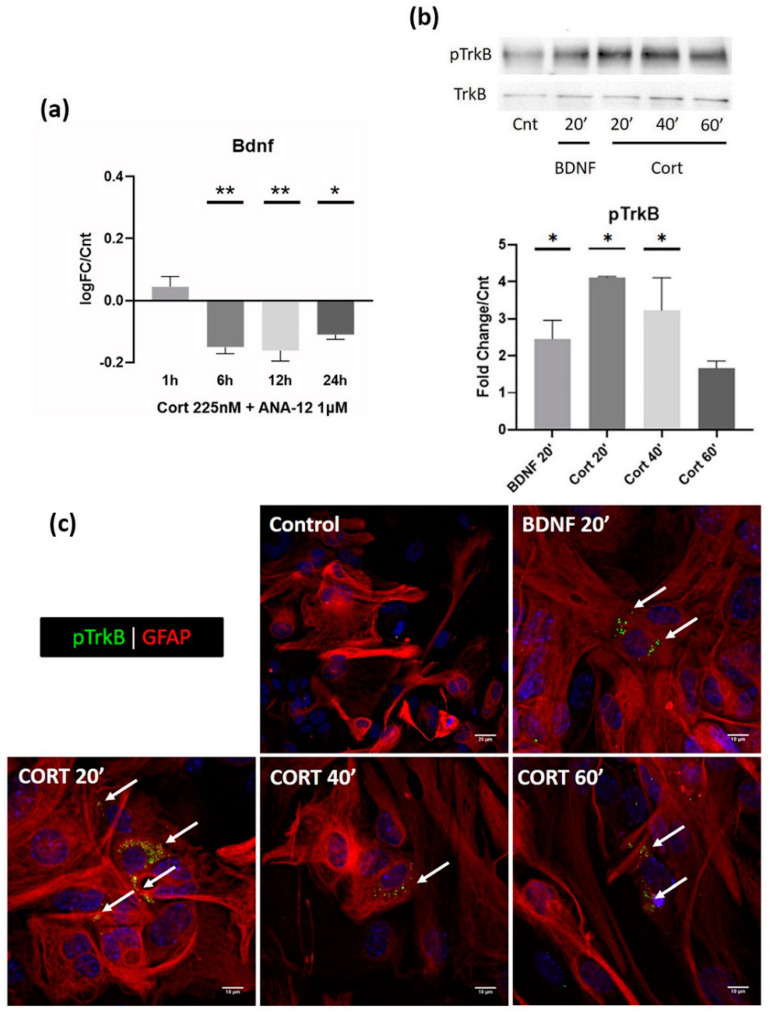
(**a**) ANA-12 (1 μΜ) administered simultaneously with Cort (225 nM) was able to completely abolish the Cort-mediated early response on astrocytes, restoring *Bdnf* expression back to its baseline control levels. At the same time, ANA-12 had no effect at all on the late response at any of the tested timepoints (6 h, 12 h and 24 h). (**b**) The phosphorylation of TrkB on astrocytes was measured after Cort administration (225 nM) for 20′, 40′ and 60′ with WB analysis. BDNF (500 ng/mL) administration for 20′ was used as a positive control of TrkB activation. Levels of phospho-TrkB are expressed as pTrkB/TrkB ratio normalized to the untreated control. Cort significantly increased pTrkB levels on astrocytes as early as 20′ after its administration. (**c**) Immunofluorescent staining of pTrkB (green) on astrocytes (GFAP, red) after Cort administration (225 nM) for 20′, 40′ and 60′ or BDNF (500 ng/mL) administration for 20′ (positive control). Phosphorylation of TrkB (arrows) was profoundly more visible both in the BDNF treatment (scale bar = 10 μm) and all the Cort treatment timepoints (scale bar = 10 μm) compared to the control condition (scale bar = 25 μm) where pTrkB stain was scarce. *n* = 3 for each experiment. * *p* < 0.05, ** *p* < 0.01; one-way ANOVA. Data are mean ± S.E.M.

**Figure 4 biomolecules-12-01322-f004:**
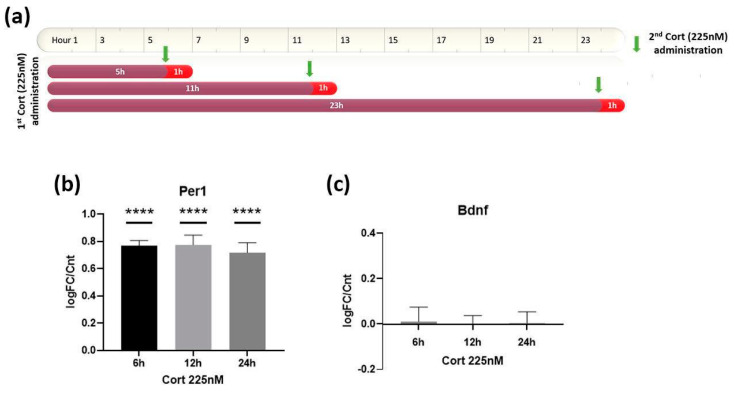
(**a**) Schematic illustration of the experimental design that was used to study the transcriptional effects of 2 Cort administrations on *Bdnf* in astrocytes. In all 3 treatments, the first dose of Cort is administered at the timepoint 0 h and then the second one exactly 1 h before the end of each treatment (5 h for the 6 h treatment, 11 h for the 12 h treatment and 23 h for the 24 h treatment). qRT-PCR was performed for RNA samples collected at the end of each timepoint. (**b**) *Per1* mRNA expression levels remained equally elevated at all 3 timepoints after the 2nd dose of Cort administration (225 nM). (**c**) The second administration of Cort restored the *Bdnf* expression levels back to the untreated control ones independently of the time delay between the two pulses. *n* = 3 for each experiment. **** *p* < 0.0001; one-way ANOVA. Data are mean ± S.E.M.

## Data Availability

All data are included in this paper and the Appendix A.

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
