# Peer review of "Biphasic Response of Astrocytic Brain-Derived Neurotrophic Factor Expression following Corticosterone Stimulation"

_biomolecules, 2022, doi:10.3390/biom12091322_

Round 1
Reviewer 1 Report
In their manuscript Tsimpolis and colleagues described the temporal regulation of BDNF expression in astrocytes by corticosterone stimulation, an interest aspect for cell physiology and also for neurological disorders.
The paper is written in a fluent English and the rationale of the study is clearly described. However, the experiments performed provide limited information regarding such regulation. I thus suggest some modifications for the improvement of this manuscript.
Major modifications:
· To molecularly support the data, I suggest to perform an experiment aimed at measuring Bdnf protein expression in astrocytes or in their culture medium.
· In addition, an immunofluorescence for p-TrkB in the membrane of stimulated astrocytes might reinforce Western blot data in Fig.2b.
· In Fig.3, the authors add the interesting information of pulsing effect of Corticosteron on Bdnf expression. However, they do not investigate the possible receptor involved. Pharmacological or molecular experiments could answer to this question.
Minor modifications:
· The nomenclature of genes and proteins is somewhere uncorrect. Please, write them in the correct way.
· In the introduction some references are lacking to support some important points. In particular, I suggest to include refs for this statement: “This pulsatility is of potential neurobiological significance, since GCs bind to and activate intracellularly two types of sensitive receptors, the glucocorticoid receptors (GRs) and the mineralocorticoid receptors (MRs), which are known to be highly expressed in neural and glial cells in many brain regions” and for the following one: “It is of special notice that GCs and their receptors are expressed in higher concentrations among all steroids in the brain, while they are considered the major regulating factor in stress, a condition where BDNF is also highly involved”.
· In the Methods section, the author need to explain why they used AraC to obtain pure astrocyte cultures.
· Further, they have to write which isoform of BDNF has been selected for qRT-PCR analysis.
· For Western blot experiments, it is not clear whether TrkB expression is analysed on the same membrane where phospho-TrkB was measured after stripping or in a different western blot.
· In Results section, in Fig.1E the authors assessed the ability of GR and MR inhibitors to prevent Per1 activation by Corticosteron. It emerges that spironolactone does not block Per1 activation and accordingly it cannot affect Bdnf expression. This could be associated with the reduced expression of MR, compared to GR but this point has not been discussed by the authors.
Author Response
We would like to thank the reviewer for her/his detailed and very helpful review on our manuscript. The provided comments were fully explored and they led to the performance of new experiments that are now added in our paper and significantly improve our work.
Major comments
- Comment 1: To molecularly support the data, I suggest to perform an experiment aimed at measuring Bdnf protein expression in astrocytes or in their culture medium.
Response: We would like to thank the reviewer for her/his valuable suggestion. We fully agree that measuring the BDNF protein levels is an important part, complementary to our previous RNA experiments. Thus, ELISA measurements were now performed on both astrocytic lysates and their culture medium that were treated with corticosterone for the same timepoints (1h, 6h, 12h and 24h) to evaluate the levels of mature BDNF, both its secreted amount and the intracellularly produced and packed quantity. Our results showed BDNF is not detected in the culture medium, perhaps due to lack of signals that induce its release, but its detection was successful in the astrocytic lysates. These results have now been included in Figure 1 and are elaborately discussed in the proper parts. Consequently, the subsection 2.7 ELISA in the Materials and Methods section was added, describing the ELISA protocol that was used, the subsection 3.1 was enriched with one more paragraph describing these results while one more paragraph was added in the Discussion section. Finally, small additions were made in the Abstract, Introduction and Conclusion text of the manuscript.
- Comment 2: In addition, an immunofluorescence for p-TrkB in the membrane of stimulated astrocytes might reinforce Western blot data in Fig.2b.
Response: We would like to also thank the reviewer for this suggestion. Immunostainings have now been performed in coverslips confluent with astrocytes upon treated with BDNF and Corticosterone at the same way to the treatments performed in the results shown in the Western Blots (previously Figure 2b, now Figure 3b). The immunofluorescent stainings are now depicted in Figure 3c, while appropriate text information has also been added in the legend of the Figure and the manuscript (sub-section 3.3). Finally, sub-section (2.6 Immunocytochemistry) was added in the 2. Materials and Methods section of the manuscript, describing the immunofluorescent protocol used as well as the necessary microscopy details and settings by which the aforementioned images were obtained.
- Comment 3: In Fig.3, the authors add the interesting information of pulsing effect of Corticosteron on Bdnf expression. However, they do not investigate the possible receptor involved. Pharmacological or molecular experiments could answer to this question.
Response: This is an excellent suggestion and it is actually one of our next aims of a recently funded project in the lab, focusing on elucidating the significance and the details of mechanism of action of glucocorticoid pulsatility in neuronal survival and neuron-glia interplay, exploring in depth the involved receptor system and the downstream mediators. This ongoing project is the continuation of the work presented in the current manuscript and its full completion will be a time-consuming process. Thus, we think that our answer on this reviewer’s comment cannot be included in the present manuscript.
Minor comments
- Comment 1: The nomenclature of genes and proteins is somewhere uncorrect. Please, write them in the correct way.
Response: We would like to apologize for some miswritten nomenclature of genes and proteins. The necessary corrections have been made in the manuscript and the nomenclature of all genes and proteins is now written correctly.
- Comment 2: In the introduction some references are lacking to support some important points. In particular, I suggest to include refs for this statement: “This pulsatility is of potential neurobiological significance, since GCs bind to and activate intracellularly two types of sensitive receptors, the glucocorticoid receptors (GRs) and the mineralocorticoid receptors (MRs), which are known to be highly expressed in neural and glial cells in many brain regions” and for the following one: “It is of special notice that GCs and their receptors are expressed in higher concentrations among all steroids in the brain, while they are considered the major regulating factor in stress, a condition where BDNF is also highly involved”.
Response: We would like to thank the reviewer for turning our attention to these two points of the introduction. We have now performed some slight rephrasing in the text of the manuscript to increase the accuracy of our arguments and we have also included the necessary references to support these parts.
- Comment 3: In the Methods section, the author need to explain why they used AraC to obtain pure astrocyte cultures.
Response: We would like to thank the reviewer for this comment and apologize for not clarifying the anti-mitotic effects of Ara-C, that play a crucial role in inhibiting the proliferation of the rapidly proliferative microglial cells and significantly purifying on this way our primary astrocytic cultures. The text in sub-section 2.2 of the manuscript, has been modified appropriately so that the reader can now immediately comprehend the important role of the use of Ara-C in our protocol.
- Comment 4: Further, they have to write which isoform of BDNF has been selected for qRT-PCR analysis.
Response: We would like to apologize for not clarifying the Bdnf isoform that our primers are detecting. The primer pair we designed for Bdnf detects areas inside the Coding Sequence (CDS) of the gene which is included in all of the characterized splicing variants. This was a reasonable decision since this study is the first one that explores Bdnf expression under glucocorticoid treatment in astrocytes, and thus we did not want to miss any isoform. Of course, in next experiments, different Bdnf isoforms will also detected. A specific comment has now been added in the sub-section 2.4 of the manuscript to clarify the ability of our designed primer pair to detect all possible splicing variants expressed.
- Comment 5: For Western blot experiments, it is not clear whether TrkB expression is analysed on the same membrane where phospho-TrkB was measured after stripping or in a different western blot.
Response: We would like to thank the reviewer for this comment. We have now added in the manuscript in sub-section 2.5 a comment explaining that total TrkB and pTrkB expression have been measured in different western blots, normalized in each blot to GAPDH and then the pTrkB/TrkB ratio was calculated.
- Comment 6: In Results section, in Fig.1E the authors assessed the ability of GR and MR inhibitors to prevent Per1 activation by Corticosteron. It emerges that spironolactone does not block Per1 activation and accordingly it cannot affect Bdnf expression. This could be associated with the reduced expression of MR, compared to GR but this point has not been discussed by the authors.
Response: We would like to thank the reviewer for this comment and apologize for not clarifying that spironolactone (MR inhibitor) inability to alter the expression levels of Per1 after Cort stimulation was an expected outcome, since regulation of the Per1 expression is highly correlated to the activation or inhibition of the GR and not the MR transcriptional machinery. For this reason, a comment about this result has now been added in the subsection 3.2 of the manuscript.
Therefore, we cannot directly correlate the lack of observable effects of spironolactone on Bdnf expression with its inability to inhibit the expression of Per1. Since there is no positive marker of MR transcriptional machinery activation -to our knowledge at least-, and the fact that Per1 is dependent on GR transcriptional activation, we are not allowed to suggest that spironolactone was able to successfully inhibit MR activation in our experiments. In addition, given the significantly higher affinity of MR for corticosterone compared to GR and subsequently, corticosterone’s ability to more effectively activate MR at low levels and GR at higher ones, it seems unlikely that the reduced expression of MRs by 4 times compared to GRs in our astrocytic cultures, would be sufficient to justify the lack of any effects in the expression of Bdnf.
Reviewer 2 Report
In this study, BNDF expression and TrkB activation after corticosterone treatment are investigated using time kinetic analysis. In addition, by using specific inhibitors, it was elucidated that the upregulation of BNDF expression after 1 hour is dependent on phosphorylation of TRKB and the downregulation after 6, 12 and 24 hours is dependent on GR activity. The study is well comprehensible and structured. The introduction adequately introduces the topic and refers very specifically to the analyses performed. The material and methods section should be commended here, as it provides an excellent understanding of the experiments performed. Overall, the manuscript contributes to the understanding of corticosterone treatment mechanisms in astrocytes and is also interpreted appropriately by the authors, leaving only a few questions unanswered. I have noticed the following points:
Minor points
Line 9, 12 and 21
There is one space too many inserted here.
Line 123-124
Please indicate where you obtained the RU-486 and ANA-12 inhibitors. Since the specificity of the inhibitors (RU-486, spironolactone, ANA-12) was not tested by the authors, it would be necessary to indicate why this inhibitor concentration was selected. Is there any previous work demonstrating inhibition at the concentration used in astrocytes? Please explain or cite the literature.
Line 119, 175-178
The authors describe that the corticosterone concentration used is the physiological murine concentration and that this concentration was determined before the start of the experiments. However, neither in the material and methods section nor in the results section is there any indication of how this concentration was determined and where the data can be found. Is there any preliminary work on this? Then it should be cited. Were the data in the supplement or were the data not shown? Please state this clearly.
Line 150, 159
The manufacturer in this context is probably ThermoFisher Scientific.
Line 161
The manufacturer of the nitrocellulose membrane is missing.
Line 162
I think here it would be better to write "or" instead of "and".
Line 174
Please delete this line.
Line 231
In my view, the manuscript would be more readable if Figure 1 e-g were separated from Figure 1 a-d. Then the arrangement of Figure 1 could be improved and the text sections of the results would correspond to the Figures.
Line 319-384 (Discussion)
Here it would be appropriate to improve the discussion of the cell system limitations.
A further point is that BNDF was determined exclusively at the mRNA level.
What exactly should further studies investigate?
If BDNF is one of the most important neuroprotective, neurogenic, and synaptogenic factors of the central nervous system, what clinically relevant hypotheses might be drawn from the results (if confirmed at the human level)?
Overall, a better discussion of the results with the existing literature would be informative here.
Author Response
Heraklion, September 09, 2022
We would like to thank the reviewer for her/his detailed and very helpful review on our manuscript. The provided comments were fully explored and they led to the performance of new experiments that are now added in our paper and significantly improve our work.
- Comment 1: Line 9, 12 and 21 - There is one space too many inserted here.
Response: As correctly pointed out by the reviewer, the extra spaces were removed.
- Comment 2: Line 123-124 - Please indicate where you obtained the RU-486 and ANA-12 inhibitors. Since the specificity of the inhibitors (RU-486, spironolactone, ANA-12) was not tested by the authors, it would be necessary to indicate why this inhibitor concentration was selected. Is there any previous work demonstrating inhibition at the concentration used in astrocytes? Please explain or cite the literature.
Response: The manufacturer and the catalogue number of all inhibitors is indicated in sub-section “2.1 Reagents” of the manuscript. Regarding the selected concentrations of the inhibitors used, we would like to apologize for not clarifying how we came down to them. Thus, the concentration of ANA-12 selected for TrkB inhibition was 1μM, as this concentration was evaluated as competent enough to completely abolish potential protective effects of BDNF on astrocytes without simultaneously eliciting astrocytic death, according to a recent publication which we have now cited in the sub-section 2.3 of the manuscript.
For the concentration of RU-486, three different concentrations (within its effective dose range) were evaluated in order to establish which one would inhibit significantly the activation of GR, an effect which was tested through the expression levels of Per1 gene. In all 3 concentrations tested, Per1 mRNA expression levels were significantly reduced, so we have chosen to show only one of them (25uM). A comment was added in the appropriate sub-section 3.2 of the manuscript to clarify that in all 3 concentrations of RU-486 the levels of Per1 mRNA were measured, but we only show the results at 25μM for illustrative purposes.
In our experiments we decided to use 2.5μM of spironolactone. Unfortunately, there isn’t any in vitro study -to our knowledge at least- that spironolactone was used for blocking glucocorticoid-mediated genomic effects. Several studies in other systems (https://pubmed.ncbi.nlm.nih.gov/16269966/; https://journals.physiology.org/doi/full/10.1152/physiolgenomics.00128.2016) have used 2-4x higher than ours concentrations of spironolactone, but they measured non-relevant effects compared to our study (like the inhibition of the genomic MR-mediated aldosterone effects on NA+/H+ exchange in vascular smooth muscle cells or gene expression effects on human skeletal muscle myoblast cultures respectively).
In other studies too (https://pubmed.ncbi.nlm.nih.gov/9231795/, https://pubmed.ncbi.nlm.nih.gov/10086974/), they used concentration of 1μM and 10μM respectively to inhibit some non-genomic effects of aldosterone, but they were unsuccessful. Thus, we think that the 2.5μM concentration is an active, non-toxic concentration that can effectively work.
- Comment 3: Line 119, 175-178 - The authors describe that the corticosterone concentration used is the physiological murine concentration and that this concentration was determined before the start of the experiments. However, neither in the material and methods section nor in the results section is there any indication of how this concentration was determined and where the data can be found. Is there any preliminary work on this? Then it should be cited. Were the data in the supplement or were the data not shown? Please state this clearly.
Response: We would like to thank the reviewer for this comment and apologize for not clarifying how this physiological corticosterone concentration (225nM) was determined. According to the existing literature, the corticosterone levels circulating in the serum and plasma of unstressed mice, has been measured to vary between 50ng/ml (or 145nM) and 300ng/ml (or 865nM). Since the aim of this work, is to clarify the effects of a physiological pulse of corticosterone (both short and long-term) on the expression of the astrocytic BDNF in vitro, we decided to use a concentration at 225nM as it would fall above the necessary concentration to physiologically activate the GR and MR receptors, but undoubtedly below the threshold of being characterized as a concentration that could imitate the corticosterone secretion levels under stress. The necessary literature has now been cited and text changes have been performed in the sub-section 3.1 of the manuscript.
- Comment 4: Line 150, 159 - The manufacturer in this context is probably ThermoFisher Scientific.
Response: As correctly pointed out by the reviewer, the name of the manufacturer was corrected.
- Comment 5: Line 161 - The manufacturer of the nitrocellulose membrane is missing.
Response: As correctly pointed out by the reviewer, the manufacturer of the nitrocellulose, along with the catalogue number of the product were added.
- Comment 6: Line 162 - I think here it would be better to write "or" instead of "and".
Response: As suggested by the reviewer, “and” was replaced by “or”.
- Comment 7: Line 174 - Please delete this line.
Response: As correctly pointed out by the reviewer, the line was deleted.
- Comment 8: Line 231 - In my view, the manuscript would be more readable if Figure 1 e-g were separated from Figure 1 a-d. Then the arrangement of Figure 1 could be improved and the text sections of the results would correspond to the Figures.
Response: As correctly pointed out by the reviewer, in order to improve the readability of the manuscript as well as for illustrative purposes, we have now separated Figure 1 e-g from Figure 1 a-d and renamed it to “Figure 2”.
- Comment 9: Line 319-384 (Discussion) - Here it would be appropriate to improve the discussion of the cell system limitations. A further point is that BNDF was determined exclusively at the mRNA level. What exactly should further studies investigate? If BDNF is one of the most important neuroprotective, neurogenic, and synaptogenic factors of the central nervous system, what clinically relevant hypotheses might be drawn from the results (if confirmed at the human level)? Overall, a better discussion of the results with the existing literature would be informative here.
Response: We would like to thank the reviewer for this comment. According to the suggestions on the comment, we have performed specifc changes in the Discussion section of the manuscript. More specifically, regarding the cell system limitations, we have added a comment explaining the limitations of our 2D, in vitro system in the interpretation of our results (e.g., astrocyte morphology, cell-to-cell interaction, interplay with other glial and neuronal cultures) and the necessity for replication of our results in different and more complex biological systems.
Regarding the point of measuring BDNF exclusively at the mRNA level, after the major revisions requested by Reviewer 1, we have now performed measurements of mature BDNF on the protein level using ELISA on our timepoint experiment. The results of this experiment can be found in sub-section 3.1 and in Figure 1. In addition, a paragraph has been added at the end of the Discussion commenting on the questions that have been aroused from this work and how future studies could tackle them, in order to further elucidate the demonstrated biphasic phenomenon and advance our knowledge on the interesting interplay between BDNF and glucocorticoids.
Finally, regarding the suggestion about the clinically relevant hypotheses that could be drawn from our results, although this is an in vitro work in mouse primary cultures that requires further research to fully unravel the underlying mechanisms and performance of respective experiments in different cell populations and more complex biological systems like human brain, we have added few lines on Discussion part about the potential therapeutic role of these findings. At this stage of research, we would not prefer to formulate any more extrapolating hypotheses about its clinical significance in humans.

Reviewer 3 Report
Comments:
This manuscript entitled " Biphasic response of astrocytic brain-derived neurotrophic factor expression following corticosterone stimulation" by Tassinari et al. is interesting. However, the techniques are not sufficient to prove the results and the evidence is not strong enough to confirm the hypothesis. The manuscript needs major revisions before it can be recommended for publication.
Author Response
We would like to thank the reviewer for her/his detailed and very helpful review on our manuscript. The provided comments were fully explored and they led to the performance of new experiments that are now added in our paper and significantly improve our work.
- Comment: This manuscript entitled " Biphasic response of astrocytic brain-derived neurotrophic factor expression following corticosterone stimulation" by Tassinari et al. is interesting. However, the techniques are not sufficient to prove the results and the evidence is not strong enough to confirm the hypothesis. The manuscript needs major revisions before it can be recommended for publication.
Response: We would like to thank the reviewer for her/his assessment. We have now significantly revised the manuscript by Tsimpolis et al, adding, correcting and rewriting experiments, text and figures and sections at Introduction, Materials and Methods and Discussion parts.
Round 2
Reviewer 1 Report
None
Reviewer 3 Report
Accept in present form